# Thermal Hall conductivity in the cuprate Mott insulators $Nd_2CuO_4$ and $Sr_2CuO_2Cl_2$

Marie-Eve Boulanger [1,6], Gaël Grissonnanche [1,6], Sven Badoux [1], Andréanne Allaire[1], Étienne Lefrançois[1], Anaëlle Legros [1,2], Adrien Gourgout[1], Maxime Dion [1], C. H. Wang[3], X. H. Chen [3], R. Liang[4], W. N. Hardy[4], D. A. Bonn[4] & Louis Taillefer [1,5 ✉]

The heat carriers responsible for the unexpectedly large thermal Hall conductivity of the cuprate Mott insulator $La_2CuO_4$ were recently shown to be phonons. However, the mechanism by which phonons in cuprates acquire chirality in a magnetic field is still unknown. Here, we report a similar thermal Hall conductivity in two cuprate Mott insulators with significantly different crystal structures and magnetic orders – $Nd_2CuO_4$ and $Sr_2CuO_2Cl_2$ – and show that two potential mechanisms can be excluded – the scattering of phonons by rare-earth impurities and by structural domains. Our comparative study further reveals that orthorhombicity, apical oxygens, the tilting of oxygen octahedra and the canting of spins out of the $CuO_2$ planes are not essential to the mechanism of chirality. Our findings point to a chiral mechanism coming from a coupling of acoustic phonons to the intrinsic excitations of the $CuO_2$ planes.

[1] Institut Quantique, Département de Physique & RQMP, Université de Sherbrooke, Sherbrooke, QC J1K 2R1, Canada. [2] SPEC, CEA, CNRS-UMR3680, Université Paris-Saclay, Gif-Sur-Yvette, France. [3] Hefei National Laboratory for Physical Science at Microscale and Department of Physics, University of Science and Technology of China, Hefei, Anhui 230026, People's Republic of China. [4] Department of Physics & Astronomy, University of British Columbia, Vancouver, BC V6T 1Z1, Canada. [5] Canadian Institute for Advanced Research, Toronto, ON M5G 1M1, Canada. [6] These authors contributed equally: Marie-Eve Boulanger, Gaël Grissonnanche. ✉email: louis.taillefer@usherbrooke.ca

In the last decade, the thermal Hall effect has become a useful probe of insulators[1], because it can reveal whether the carriers of heat in a material have chirality. (Here we use the term "chirality" to mean handedness in the presence of a magnetic field.) In insulators, the carriers of heat are not charged, but neutral, and so the electrical Hall effect is zero. The thermal Hall conductivity $\kappa_{xy}$ is measured by sending a heat current along the $x$-axis and detecting a transverse temperature gradient along the $y$-axis, in the presence of a perpendicular magnetic field (along the $z$-axis). It has been shown that in certain conditions, spins can produce such chirality[2]. For example, magnons give rise to a thermal Hall signal in the antiferromagnet $Lu_2V_2O_7$ (ref. [3]). As a result, a measurement of the thermal Hall effect can in principle provide access to various topological excitations in insulating quantum materials, such as Majorana edge modes in chiral spin liquids[4]. Recently, the thermal Hall conductivity $\kappa_{xy}$ seen in α-$RuCl_3$ below $T \simeq 80\,K$ (refs. [5,6]) has been attributed to the excitations of a Kitaev spin liquid[7]. Similarly, the $\kappa_{xy}$ signal observed in some frustrated magnets—in which there is no magnetic order down to the lowest temperatures—has been attributed to spin-related heat carriers[8].

However, phonons can also generate a nonzero thermal Hall conductivity if some mechanism confers chirality to them. For instance, an intrinsic mechanism is the Berry curvature of phonon bands acquired from a magnetic environment[9]. In the ferrimagnetic insulator $Fe_2Mo_3O_8$, the large $\kappa_{xy}$ signal is attributed to the strong spin–lattice coupling characteristic of multiferroic materials[10]. An extrinsic mechanism is the skew scattering of phonons by rare-earth impurities[11], as in the rare-earth garnet $Tb_3Ga_5O_{12}$ (refs. [12,13]). Recently, a large phononic $\kappa_{xy}$ has been observed in the nonmagnetic insulator $SrTiO_3$ (ref. [14]). A proposed explanation involves the large ferroelectric susceptibility of this oxide insulator, together with an extrinsic mechanism whereby phonons are scattered by the polar boundaries from the antiferrodistortive structural transition at 105 K (ref. [15]). This interpretation is supported by the fact that $\kappa_{xy}$ is negligible in the closely related material $KTaO_3$ (ref. [14]), which remains cubic and free of structural domains.

In cuprates, a large negative $\kappa_{xy}$ signal was observed at low temperature inside the pseudogap phase[16], i.e., for dopings $p < p^*$, where $p^*$ is the pseudogap critical doping[17]. Because it persists down to $p = 0$, in the Mott insulator state, this negative $\kappa_{xy}$ cannot come from charge carriers, which are not mobile at $p = 0$. Therefore, it must come either from spin-related excitations (possibly topological, as in refs. [18,19]) or from phonons (as in ref. [15]). To distinguish between these two types of heat carriers, a simple approach was recently adopted: the thermal Hall conductivity was measured for a heat current along the $c$-axis, normal to the $CuO_2$ planes, a direction in which only phonons move easily[20]. In $La_2CuO_4$, at $p = 0$, the thermal Hall signal was found to be just as large as for an in-plane heat current, i.e., $\kappa_{zy}(T) \approx \kappa_{xy}(T)$ (ref. [21]). This is compelling evidence that phonons are the heat carriers involved in the thermal Hall signal of this insulator. Moreover, it was found that this phonon Hall effect vanishes entirely immediately outside the pseudogap phase, i.e., $\kappa_{zy}(T) = 0$ at $p > p^*$, revealing that phonons only become chiral upon entering the pseudogap phase[21].

The question is: what makes the phonons in cuprates become chiral? In order to provide answers to this question, we have investigated two other cuprate Mott insulators, $Nd_2CuO_4$ and $Sr_2CuO_2Cl_2$, and find in both a large negative thermal Hall conductivity similar to that of $La_2CuO_4$. While the three materials share the same fundamental characteristic of cuprates, namely they are a stack of single $CuO_2$ planes, there are significant differences between them (see "Methods" and Fig. 1). Our comparative study allows us to conclude that none of the

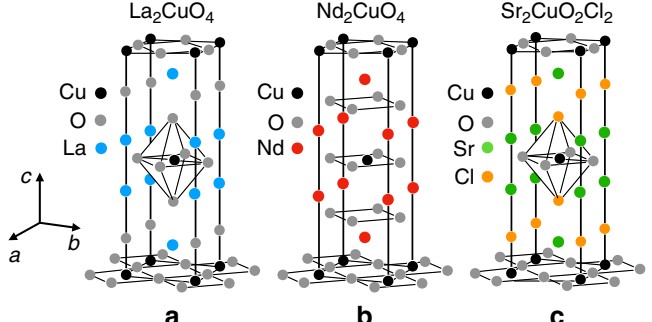

**Fig. 1 Crystal structure of $La_2CuO_4$, $Nd_2CuO_4$, and $Sr_2CuO_2Cl_2$.** Sketch of the crystal structure of the three single-layer cuprate Mott insulators compared in the present study: **a** $La_2CuO_4$, **b** $Nd_2CuO_4$, and **c** $Sr_2CuO_2Cl_2$. Note that the small orthorhombic distortion in $La_2CuO_4$ below 530 K is not shown here, nor is the tilt in the oxygen octahedra surrounding the Cu atoms.

distinguishing features—orthorhombicity, structural domain boundaries, apical oxygens, spin canting, noncollinear alignment of spins, and nature of the cation—play a key role in causing the chirality. This points to a chiral mechanism associated with the coupling of phonons to the $CuO_2$ planes themselves.

## Results

**Thermal Hall conductivity.** In Fig. 2, we show our data for $\kappa_{xx}$ and $\kappa_{xy}$ in $Sr_2CuO_2Cl_2$ (sample A) and $Nd_2CuO_4$. We see that as in $La_2CuO_4$, both materials show a large negative thermal Hall signal. We also observe a certain field dependence of $\kappa_{xx}$, larger than the small one observed in $La_2CuO_4$ (ref. [16]). In $Sr_2CuO_2Cl_2$, the field increases $\kappa_{xx}$ slightly, below $T \approx 20\,K$. In $Nd_2CuO_4$, the field decreases $\kappa_{xx}$, below $T \approx 40\,K$.

In Fig. 3, we compare the three cuprate Mott insulators. We observe that the curves of $-\kappa_{xy}$ vs. $T$ (right panels) are similar in shape, peaking at $T \approx 25\,K$, a temperature close to that where $\kappa_{xx}$ vs. $T$ peaks (left panels). At low temperature, $\kappa_{xx}$ is dominated by phonons. Indeed, because there is a gap in the magnon spectrum of these antiferromagnets[22], their contribution to $\kappa_{xx}$ becomes negligible at low $T$ compared to the phonon contribution. At $T = 35\,K$, the magnon conductivity is only 2% of the measured $\kappa_{xx}$ (ref. [20]), and it rapidly becomes vanishingly small below that temperature. At $T = 20\,K$, the magnitude of $\kappa_{xx}$ is 8 times larger in $Nd_2CuO_4$ compared to $Sr_2CuO_2Cl_2$ (Fig. 3). So, phonons are a lot more conductive in $Nd_2CuO_4$. We see from Fig. 3e, f that $\kappa_{xy}$ is correspondingly (ten times) larger in $Nd_2CuO_4$. This is strong evidence that phonons are the heat carriers responsible for the Hall response.

In Fig. 4, we plot the ratio $\kappa_{xy}/\kappa_{xx}$ vs $T$ for the three materials. We see that not only is this ratio of similar magnitude in the three cuprates, but its temperature dependence is also very similar, growing with decreasing $T$ to reach a maximal (negative) value at $T \approx 10$–15 K, where $|\kappa_{xy}/\kappa_{xx}| \approx 0.3$–0.4% (at $H = 15\,T$).

Having observed a large negative thermal Hall conductivity $\kappa_{xy}$ in both $Nd_2CuO_4$ and $Sr_2CuO_2Cl_2$ that is very similar to that previously reported for $La_2CuO_4$ (i.e., of comparable magnitude when measured relative to $\kappa_{xx}$) allows us to draw several conclusions about the underlying mechanism for chirality in the cuprate Mott insulators.

**Spin canting.** It has been shown theoretically that in ferromagnetic or antiferromagnetic insulators, under certain conditions, magnons can have chirality and should give rise to a thermal Hall effect[1]. In the collinear Néel antiferromagnetic order of $La_2CuO_4$, no thermal Hall effect is expected theoretically, because of the so-

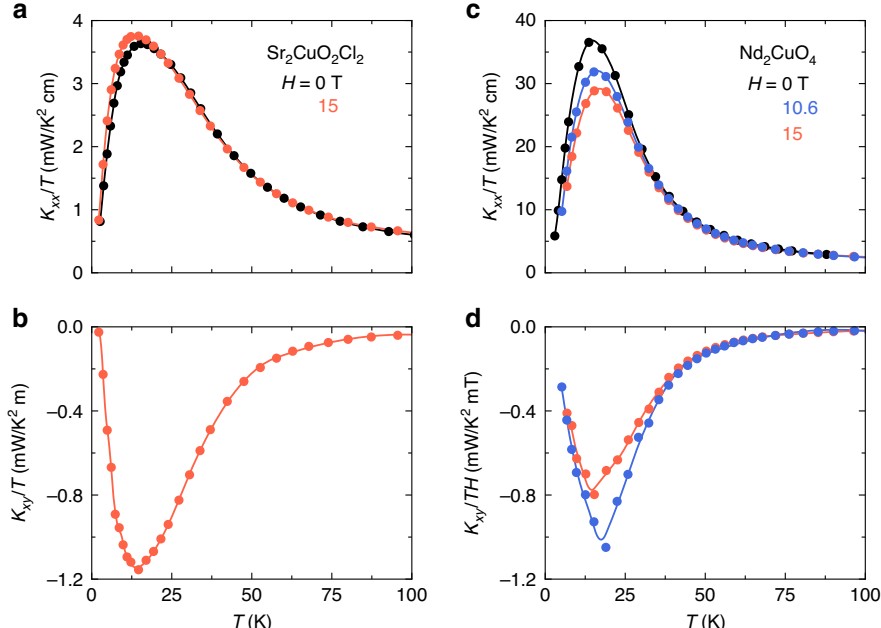

**Fig. 2 Thermal transport in Sr$_2$CuO$_2$Cl$_2$ and Nd$_2$CuO$_4$. a** Thermal conductivity of Sr$_2$CuO$_2$Cl$_2$ (sample A) in zero field ($H = 0$, black) and in a field of 15 T applied parallel to the $c$-axis (light red), plotted as $\kappa_{xx}/T$ vs $T$. The field is seen to increase $\kappa_{xx}$ slightly at low temperature. **b** Thermal Hall conductivity of Sr$_2$CuO$_2$Cl$_2$ (same sample) in a field of 15 T applied parallel to the $c$-axis, plotted as $\kappa_{xy}/T$ vs $T$. **c** Thermal conductivity of Nd$_2$CuO$_4$, plotted as $\kappa_{xx}/T$ vs $T$, for three values of the magnetic field applied parallel to the $c$-axis: $H = 0$ (black), $H = 10.6$ T (blue), and $H = 15$ T (light red). In this case, the field is seen to decrease $\kappa_{xx}$ at low temperature. **d** Thermal Hall conductivity of Nd$_2$CuO$_4$, plotted as $\kappa_{xy}/(TH)$ vs $T$, for two values of the magnetic field applied parallel to the $c$-axis: $H = 10.6$ T (blue); $H = 15$ T (light red). The Hall conductivity $\kappa_{xy}$ is seen to be sublinear in $H$ at low $T$ and linear in $H$ at high $T$ ($T > 50$ K). All lines are a guide to the eye.

called "no-go" theorem, which states that Néel order on a square lattice has zero chirality[1]. However, if the spins of the Néel order cant out of the plane, as they do in La$_2$CuO$_4$, due to some Dzyaloshinskii–Moriya (DM) interaction, then some chirality becomes possible. In this case, one could get a nonzero $\kappa_{xy}$ signal, but it is expected to be much smaller than the measured $\kappa_{xy}$ signal in La$_2$CuO$_4$ (ref. [23]). Our data on Nd$_2$CuO$_4$ and Sr$_2$CuO$_2$Cl$_2$ completely eliminate this possibility, because a $\kappa_{xy}$ signal of similar or larger magnitude is found in these materials for which there is no canting of spins out of the plane (see "Methods"), and so no DM interaction. We conclude that magnons are not responsible for the thermal Hall effect in cuprate Mott insulators. This conclusion is consistent with the fact that in La$_2$CuO$_4$ a large $\kappa_{xy}$ signal persists down to temperatures well below the smallest magnon gap, of magnitude 26 K (ref. [22]), and up in doping well above the critical doping for the suppression of Néel order, i.e., $p \approx 0.02$ in La$_{2-x}$Sr$_x$CuO$_4$ (ref. [16]).

As for a phonon scenario whereby phonons would acquire chirality through their coupling to spins, spin canting also appears to be unimportant.

Let us now consider two mechanisms known to confer chirality to phonons in other materials—skew scattering off rare-earth impurities and scattering off structural domain boundaries—and show that neither is relevant to cuprates.

**Nature of cation**. The initial observation of a phonon thermal Hall effect, in the garnet Tb$_3$Ga$_5$O$_{12}$ (refs. [12,13]), has been attributed to the skew scattering of phonons by super-stoichiometric Tb$^{3+}$ ions[11]. This extrinsic mechanism depends crucially on the details of the crystal-field levels of the rare-earth ion. A different rare-earth ion will in general produce skew scattering of a very different strength. The fact that the ratio $\kappa_{xy}/$

$\kappa_{xx}$ is the same in all three cuprates considered here is compelling evidence that the underlying mechanism does not depend on the nature of the particular cation, whether La, Sr, or Nd.

Note also that strong skew scattering by rare-earth impurities shows up as a major reduction in $\kappa_{xx}$ (ref. [24]). In Tb$_3$Ga$_5$O$_{12}$, 2% of Tb$^{3+}$ impurities gives rise to both a finite $\kappa_{xy}$ signal from phonons (whose magnitude is given in Table 1) and a fivefold reduction in $\kappa_{xx}$ (ref. [13]), whose value at $T = 15$ K is then only $\kappa_{xx} = 1.2$ W/Km (ref. [11]). In the pyrochlore oxide Tb$_2$Ti$_2$O$_7$, a frustrated magnet with a sizable thermal Hall effect[8] (Table 1), $\kappa_{xx}$ is massively reduced compared to Y$_2$Ti$_2$O$_7$, by a factor 15 at $T = 15$ K ($H = 0$) (ref. [25]), pointing again to strong scattering of phonons by Tb$^{3+}$ ions. (Note that the thermal Hall effect in Tb$_2$Ti$_2$O$_7$ has recently been attributed to phonons[26].) By comparison, the thermal conductivity in the cuprate Mott insulators is an order of magnitude larger (Table 1), evidence that no strong skew scattering is at play: $\kappa_{xx} = 10$ W/Km in La$_2$CuO$_4$, 45 W/Km in Nd$_2$CuO$_4$, and 6 W/Km in Sr$_2$CuO$_2$Cl$_2$, at $T = 15$ K ($H = 15$ T, Fig. 3). We conclude that skew scattering of phonons by superstoichiometric cation atoms is not the mechanism that confers chirality to phonons in cuprates.

**Structural domains**. In the nonmagnetic insulator SrTiO$_3$, a negative thermal Hall conductivity was recently observed[14], with a magnitude comparable to that of the three cuprate Mott insulators (Table 1). There is little doubt that the thermal Hall effect in SrTiO$_3$ is due to phonons. Importantly, the $\kappa_{xy}$ signal in the closely related oxide KTaO$_3$ is 30 times smaller (and of opposite sign)[14] (Table 1). The key difference between the two materials is that SrTiO$_3$ undergoes an antiferrodistortive structural transition at 105 K, whereas KTaO$_3$ remains cubic down to $T \approx 0$ K. The authors of the study on those two materials conclude that the

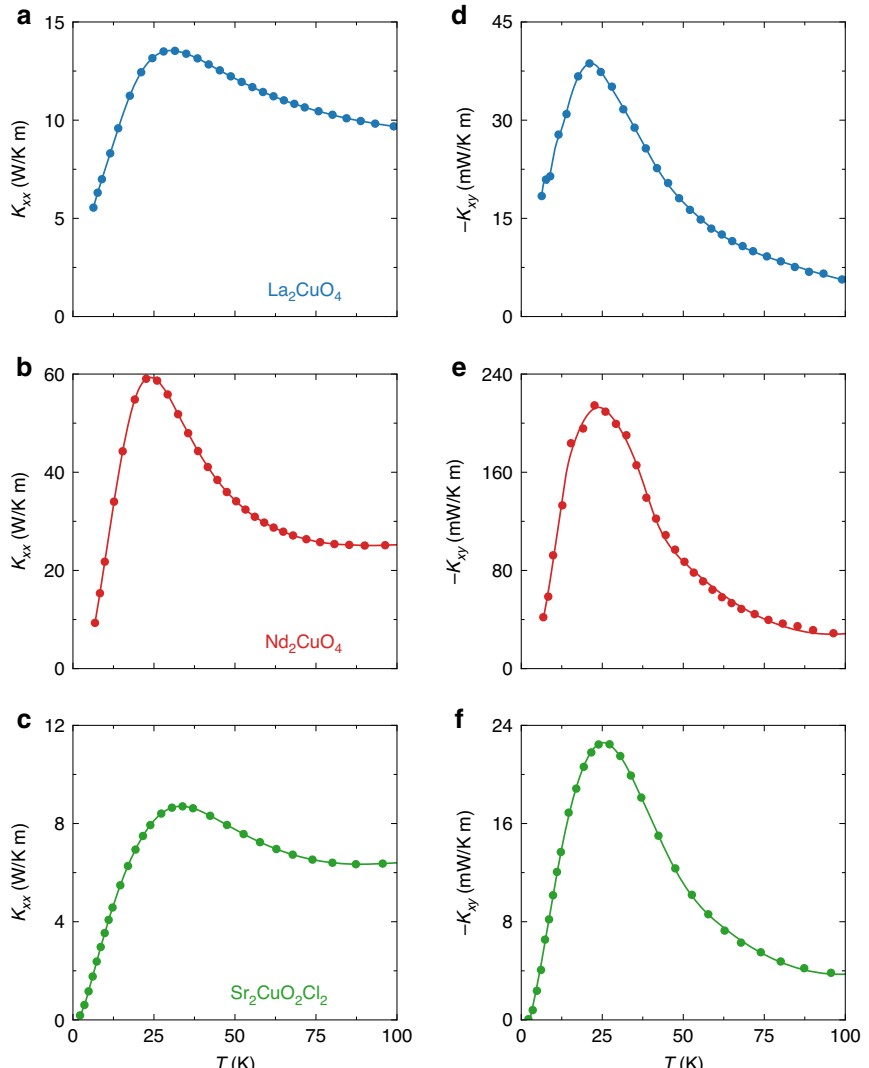

**Fig. 3 Thermal Hall conductivity in the three Mott insulators.** Left panels: thermal conductivity of the three cuprate Mott insulators, plotted as $\kappa_{xx}$ vs $T$: **a** La$_2$CuO$_4$, **b** Nd$_2$CuO$_4$, and **c** Sr$_2$CuO$_2$Cl$_2$. Right panels: Corresponding thermal Hall conductivity, plotted as $-\kappa_{xy}$ vs $T$: **d** La$_2$CuO$_4$, **e** Nd$_2$CuO$_4$, and **f** Sr$_2$CuO$_2$Cl$_2$. All data shown in this figure are taken in a field of 15 T (along the $c$-axis).

large signal in SrTiO$_3$ is linked to the structural domain boundaries that exist below 105 K (ref. [14]), although the precise mechanism whereby these confer chirality to phonons is still unclear. Our comparative study of the three cuprates allows us to rule out a similar role for structural domains. Indeed, whereas La$_2$CuO$_4$ undergoes a structural transition to an orthorhombic phase below 530 K, both Nd$_2$CuO$_4$ and Sr$_2$CuO$_2$Cl$_2$ remain tetragonal down to $T \approx 0$ K, and yet all three have a similar thermal Hall effect, in both $T$ dependence (Fig. 3) and magnitude—relative to $\kappa_{xx}$ (Fig. 4 and Table 1).

**Magnetostructural domains**. Because the collinear spin order in Sr$_2$CuO$_2$Cl$_2$ breaks the fourfold symmetry of the lattice, there will be antiferromagnetic domains below $T_N$ and these will in principle be accompanied by an orthorhombic distortion of the tetragonal lattice aligned with the moment direction in each domain. To investigate the possible effect of these putative structural distortions, we have measured $\kappa_{xy}$ in the same sample of Sr$_2$CuO$_2$Cl$_2$ (sample B) under three different conditions: (1) for a field $H = 10.6$ T applied along the $c$-axis, (2) for a field $H = 15$ T applied at an angle of 45° from the $c$-axis (whose

components normal and parallel to the CuO$_2$ planes are both 10.6 T), applied at $T = 2$ K (zero-field cooling), and (3) same as for (2), but applied at $T = 300$ K $> T_N$ (in-field cooling). In the latter in-field cooling condition, the in-plane component of the field (of magnitude 10.6 T) applied at $T > T_N$ will ensure that a single antiferromagnetic domain is present below $T_N$. (We expect the in-plane field needed to create a monodomain to be approximately 5 T, as verified in YBa$_2$Cu$_3$O$_6$ (ref. [27]).) Comparing conditions (2) and (3) amounts to comparing a multidomain sample vs a monodomain sample.

The results of this comparative study are displayed in Fig. 5. We see that $\kappa_{xy}$ is identical in the three situations, within error bars. So, magnetic domains in Sr$_2$CuO$_2$Cl$_2$, and any associated structural distortions, do not influence the thermal Hall response. Note that the noncollinear order in Nd$_2$CuO$_4$ does not break the fourfold symmetry of the lattice, so here no magnetic domains are expected.

We conclude that structural (or magnetostructural) domains are not the mechanism that confers chirality to phonons in cuprates. Moreover, the thermal Hall conductivity of cuprates is independent of whether the system has orthorhombic or

tetragonal symmetry, or whether there are apical oxygens in the structure or not.

In summary, our results show that the cuprate Mott insulators $Nd_2CuO_4$ and $Sr_2CuO_2Cl_2$ exhibit a large negative thermal Hall conductivity $\kappa_{xy}$ very similar to that found in $La_2CuO_4$. The fact that the magnitude of $\kappa_{xy}$ scales with the magnitude of the phonon-dominated $\kappa_{xx}$ as the latter varies by a factor 10 between $Sr_2CuO_2Cl_2$ and $Nd_2CuO_4$ is further evidence in favor of phonons as the carriers of heat responsible for the thermal Hall effect in these materials. Given the different crystal structures and cations involved in those three materials, the similarity in $\kappa_{xy}/\kappa_{xx}$ allows us to rule out two extrinsic mechanisms of phonon

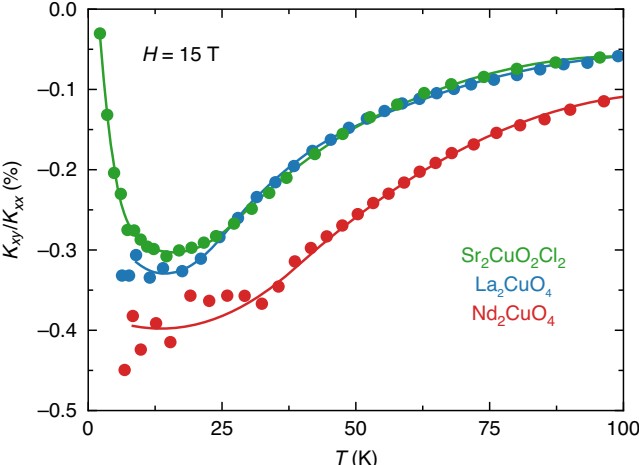

**Fig. 4 Ratio of $\kappa_{xy}$ over $\kappa_{xx}$.** Ratio of $\kappa_{xy}$ over $\kappa_{xx}$ in the three cuprate Mott insulators (expressed in %), measured in a field of 15 T applied parallel to the c-axis: $Sr_2CuO_2Cl_2$ (green), $La_2CuO_4$ (blue), and $Nd_2CuO_4$ (red). All lines are a guide to the eye. We see that despite a factor 10 in the magnitude of $\kappa_{xy}$ between $Sr_2CuO_2Cl_2$ and $Nd_2CuO_4$ (Fig. 2), the ratio $\kappa_{xy}/\kappa_{xx}$ is very similar in magnitude for all three cuprates.

chirality proposed for other oxides, namely the scattering off rare-earth impurities—invoked for $Tb_3Gd_5O_{12}$—and the scattering off structural domain boundaries—invoked for $SrTiO_3$. This suggests that phonon chirality in the cuprates comes from an intrinsic coupling of phonons to their environment.

## Discussion

Phonons can acquire chirality through a coupling to their intrinsic environment (see, e.g., ref. [9]). This could involve a coupling to charge or a coupling to spin, for example. In ref. [15], a flexoelectric coupling of phonons to their charge environment was shown to generate a Hall response. However, even in the nearly ferroelectric insulator $SrTiO_3$, where the electric polarizability is exceptionally large, this intrinsic mechanism is estimated to be much too small. The inclusion of some additional, extrinsic, scattering mechanism—possibly structural domain boundaries—is deemed necessary. Applied to cuprates, the intrinsic flexoelectric coupling is certainly much too small. It is not clear what extrinsic mechanism could be added to make this mechanism strong enough to account for the observed data in the cuprate Mott insulators.

In multiferroic materials like $Fe_2Mo_3O_8$, a large $\kappa_{xy}$ signal is observed even in the paramagnetic phase[10], where $\kappa_{xy}/\kappa_{xx} \simeq 0.5\%$ (at $T = 65$ K and $H = 14$ T) (Table 1). This is attributed to a strong spin–lattice coupling. In cuprates, a coupling of phonons to spins in their environment should be investigated as a possible source of chirality.

Another avenue of investigation for cuprates is the possibility that they harbour exotic chiral excitations, like spinons[18,19] that could couple to phonons. Such a coupling has recently been considered for the case of Majorana fermions in a Kitaev spin liquid[28].

In a scenario of phonons coupled to their environment, there could be two relevant regimes of temperature, namely above and below the peak in $\kappa_{xy}$ vs. $T$, so roughly above 25 K and below 15 K, respectively (Fig. 3). At temperatures above the peak, it has been shown that if the heat carriers have Berry curvature, they

### Table 1 Thermal Hall conductivity in various oxide insulators.

| Material | Doping | $\kappa_{xy}$ (mW/Km) | $\kappa_{xx}$ (W/Km) | $\|\kappa_{xy}/\kappa_{xx}\|$ (%) | T (K) | H (T) | Reference |
|---|---|---|---|---|---|---|---|
| $Nd_2CuO_4$ | 0.00 | −212.5 | 58.3 | 0.37 | 22 | 15 | This work |
| $Sr_2CuO_2Cl_2$ | 0.00 | −22.3 | 8.2 | 0.26 | 25 | 15 | This work |
| $La_2CuO_4$ | 0.00 | −38.6 | 12.4 | 0.30 | 20 | 12 | 16 |
| $La_2CuO_4$ (J // c) | 0.00 | −30.0 | 16 | 0.2 | 20 | 15 | 21 |
| LSCO | 0.06 | −30.0 | 5.1 | 0.58 | 15 | 15 | 16 |
| Eu-LSCO | 0.08 | −13.2 | 4.5 | 0.29 | 15 | 15 | 16 |
| Nd-LSCO (J // c) | 0.21 | −14.0 | 2.9 | 0.48 | 20 | 15 | 21 |
| Nd-LSCO (J // c) | 0.24 | 0 | 1.2 | 0 | 20 | 15 | 21 |
| Eu-LSCO (J // c) | 0.24 | 0 | 1.2 | 0 | 20 | 15 | 21 |
| $Lu_2V_2O_7$ | | +1.0 | – | 0.14 | 50 | 0.1 | 3 |
| $Tb_3Ga_5O_{12}$ | | +0.02 [a] | 0.2 | 0.01 | 5 | 3 | 13 |
| $Tb_2Ti_2O_7$ | | +1.2 | 0.27 | 0.44 | 15 | 12 | 8 |
| $Y_2Ti_2O_7$ | | 0 | 18 | 0 | 15 | 8 | 8,25 |
| $(Tb_{0.3}Y_{0.7})_2Ti_2O_7$ | | +3.8 | 1.0 | 0.38 | 15 | 12 | 26 |
| $SrTiO_3$ | | −80 | 36 | 0.20 | 20 | 12 | 14 |
| $KTaO_3$ | | +2 | 30 | 0.007 | 30 | 12 | 14 |
| $Fe_2Mo_3O_8$ | | +12 | 2.5 | 0.48 | 65 | 14 | 10 |

[a]Expected to be 10 times larger at $T = 20$ K and $H = 15$ T. The magnitude and sign of $\kappa_{xy}$ are given for a temperature $T$ and magnetic field $H$ as indicated. The quoted values are typically the largest absolute values for a field of 15 T or so. The value of $\kappa_{xx}$ at the same $T$ and $H$ is also given, as is the corresponding ratio $|\kappa_{xy}/\kappa_{xx}|$. The first group of materials is cuprates, including the three undoped Mott insulators studied here (top) and some hole-doped cuprates, whose doping $p$ is indicated in the second column. At high doping ($p > 0.2$), the samples are not insulating but metallic, and so we quote here the thermal transport coefficients for a heat current normal to the $CuO_2$ planes (J//c), which contain only the phonon contribution to heat transport. The second group consists of one material, the ferromagnet $Lu_2V_2O_7$, whose $\kappa_{xy}$ signal is due to magnons. The third group consists of insulating materials with no magnetic order. It includes four pyrochlore oxides with Tb and/or Y ions, whose magnetism is either frustrated (Tb) or absent (Y), and two nonmagnetic oxides ($SrTiO_3$ and $KTaO_3$).
The last group consists of the multiferroic material $Fe_2Mo_3O_8$, which has ferrimagnetic order below 45 K. Here we quote values above that temperature, in the paramagnetic state at 65 K.

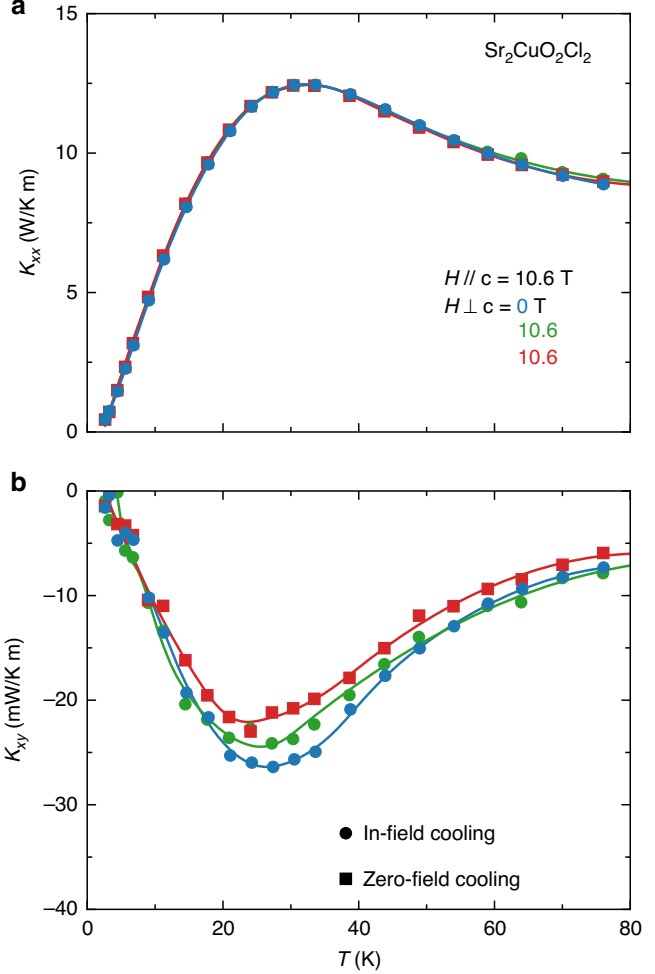

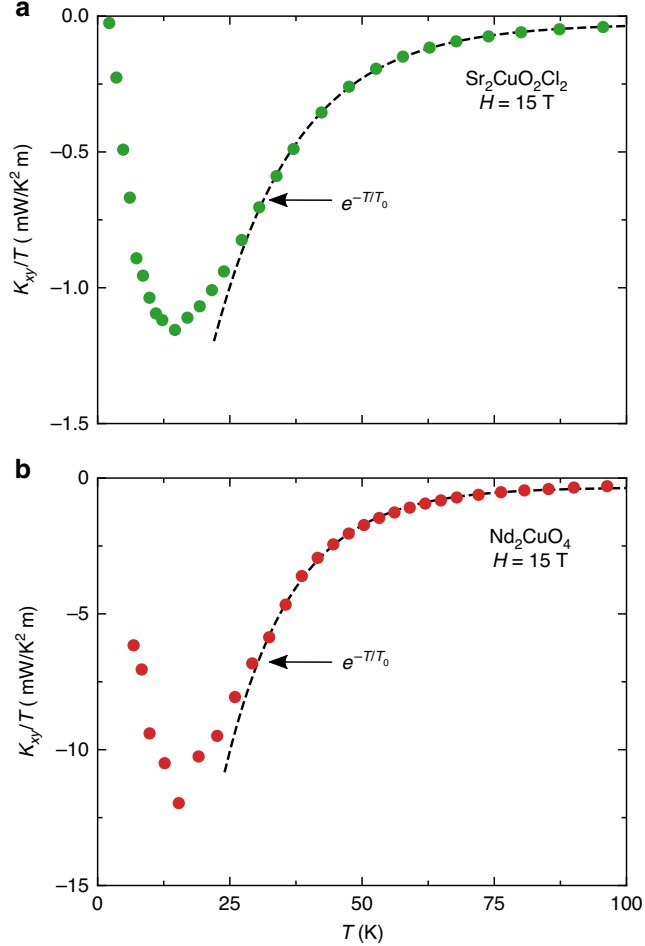

**Fig. 5 Effect of magnetostructural domains in Sr$_2$CuO$_2$Cl$_2$.** Thermal transport in Sr$_2$CuO$_2$Cl$_2$ (sample B) for a heat current $J//a$, measured as a function of increasing temperature from $T = 2$ K up to 80 K, in three different conditions: (1) for a field $H = 10.6$ T along the $c$-axis, applied at $T = 300$ K (blue circles), (2) for a field $H = 15$ T at 45° from the $c$-axis (meaning equal in-plane and out-of-plane fields, i.e., $H \perp c = H // c = 10.6$ T), applied at $T = 2$ K (zero-field cooling, red squares), and (3) for a field $H = 15$ T at 45° from the $c$-axis, applied at $T = 300$ K (in-field cooling, green circles). All lines are a guide to the eye. In conditions (1) and (2), we expect that multiple orthorhombic magnetostructural domains and associated boundaries exist below $T_N = 270$ K. Condition (3)—in-field cooling in the presence of an in-plane field of 10.6 T—ensures that a single antiferromagnetic domain exists when magnetic order sets in below $T_N$, and so there should be no, or very few, structural domain boundaries in that case. **a** Thermal conductivity $\kappa_{xx}$ vs $T$. There is no detectable difference between the three curves, showing that the presence of magnetostructural domains has a negligible impact on the phonon thermal conductivity. **b** Thermal Hall conductivity $\kappa_{xy}$ vs. $T$. Within error bars, there is no significant difference between the three curves, demonstrating that magnetostructural domains do not play a significant role in generating the thermal Hall signal in Sr$_2$CuO$_2$Cl$_2$.

**Fig. 6 Phenomenological fit to the phonon thermal Hall conductivity.** Thermal Hall conductivity, plotted as $\kappa_{xy}/T$ vs $T$ in **a** Sr$_2$CuO$_2$Cl$_2$ (sample A) and **b** Nd$_2$CuO$_4$. The data are fit to the phenomenological expression $\kappa_{xy}/T = A \exp(-T/T_0) + C$ from ref. [29]. The fit interval is 30–100 K. The resulting fit parameters are **a** $A = -5$ mW/K$^2$m, $C = -0.03$ mW/K$^2$m, $T_0 = 16$ K; **b** $A = -67$ mW/K$^2$m, $C = -0.3$ mW/K$^2$m, $T_0 = 12$ K.

would be expected to exhibit a characteristic exponential dependence, namely $\kappa_{xy}/T \propto \exp(-T/T_0)$ (ref. [29]). In Fig. 6, we fit our data on Sr$_2$CuO$_2$Cl$_2$ and Nd$_2$CuO$_4$ to that form and find a good fit over the intermediate temperature range from 30 to 100 K. An equally good fit is found for La$_2$CuO$_4$ (ref. [21]). Whether this implies that phonons acquire a Berry curvature through their coupling to the environment remains to be determined. At low temperature, we would expect phonons to eventually decouple from their environment, whether that be spins or other excitations of electronic origin. The temperature below which they do so would shed light on the nature of that coupling. In Fig. 4, we see that upon cooling below 10 K, $|\kappa_{xy}|$ in Sr$_2$CuO$_2$Cl$_2$ falls more rapidly to zero than $\kappa_{xx}$ does. (Our current data on Nd$_2$CuO$_4$ and La$_2$CuO$_4$ do not allow us to explore their regime below 10 K.) In Fig. 7, we zoom on the low-$T$ regime in Sr$_2$CuO$_2$Cl$_2$. We see that whereas $\kappa_{xx}/T^3$ rises monotonically as $T \to 0$, $\kappa_{xy}/T^3$ drops rapidly toward zero, starting roughly at 5 K. We identify 5 K as the approximate decoupling temperature between acoustic phonons and their chiral environment.

It is instructive to compare our data on undoped cuprates to prior data on hole-doped cuprates. At a doping $p = 0.24$, in both Nd-LSCO and Eu-LSCO, the thermal Hall signal coming from phonons—as opposed to charged carriers—is zero[21] (Table 1). It only becomes nonzero when the doping is reduced below the critical doping for the pseudogap phase, i.e., when $p < p^*$ ($p^* = 0.23$). The magnitude of $\kappa_{xy}$ is relatively constant from $p^*$ down to $p = 0$ when measured relative to $\kappa_{xx}$ (see Table 1), and the sign is negative throughout. This continuity suggests that the same chiral mechanism is at play in the Mott insulator and within the pseudogap phase.

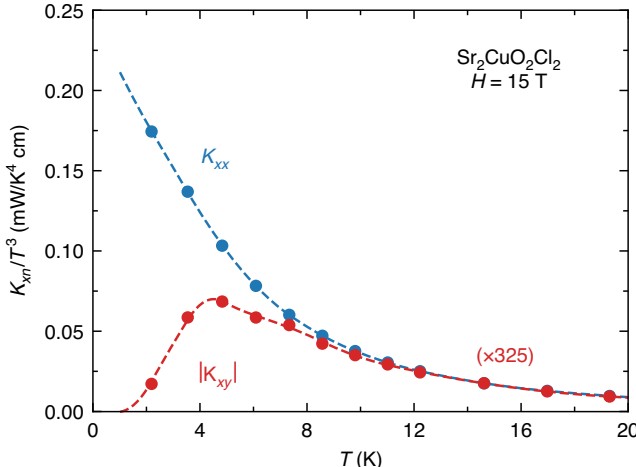

**Fig. 7 Low-temperature regime in Sr₂CuO₂Cl₂.** Thermal conductivity $\kappa_{xx}$ and thermal Hall conductivity $\kappa_{xy}$ of Sr₂CuO₂Cl₂ (sample A), plotted as $\kappa_{xx}/T^3$ (blue) and $-\kappa_{xy}/T^3$ (red) vs $T$. The dashed lines are a guide to the eye. Note that the $\kappa_{xy}$ data are multiplied by a factor 325, to compare them more easily with $\kappa_{xx}$, using only one y-axis scale.

Moreover, because the phononic $\kappa_{xy}$ signal in Nd-LSCO goes from zero at $p = 0.24$ to its full value at $p = 0.21$, rising abruptly upon crossing below $p^*$, this chiral mechanism must be an intrinsic property of the pseudogap phase—since there is no change in the crystal structure[21,30] and little change in the amount of impurity scattering between $p = 0.24$ and $p = 0.21$. This is consistent with our finding that structural domains and cation impurities are unimportant, and it extends the argument to all other defects and impurities, e.g., oxygen vacancies, all of which are essentially unchanged between $p = 0.24$ and $p = 0.21$. (In a scenario of oxygen vacancies screened by mobile charge carriers, we would expect zero screening at $p = 0$, in the insulator, so we should see a much larger $\kappa_{zy}$ signal in La₂CuO₄ than in Nd-LSCO at $p = 0.21$, for example. This is not the case, on the contrary. As seen from Table 1, the $\kappa_{zy}$ signal is larger at $p = 0.21$, relative to $\kappa_{zz}$: $|\kappa_{zy}/\kappa_{zz}| = 0.48\%$ in Nd-LSCO $p = 0.21$ vs. 0.2% in La₂CuO₄.)

One intrinsic mechanism has recently been proposed whereby phonons couple to an electronic state that breaks time-reversal and inversion symmetries, which would be realized in the pseudogap phase of cuprates[31].

Another possible mechanism is the coupling of phonons to short-range antiferromagnetic correlations. Experimental evidence for such correlations includes the Fermi-surface transformation across $p^*$ observed by angle-dependent magnetoresistance[32] and the drop in carrier density across $p^*$ observed in the electrical Hall effect[33–35], both consistent with spin modulations with a wave-vector $Q = (\pi, \pi)$. Solutions of the Hubbard model in the paramagnetic state find that, in doped Mott insulators such as the cuprates, local moments[36,37] persist all the way from half-filling up to a critical doping where the pseudogap disappears[38–40]. In calculations within the pseudogap phase, superexchange between local moments naturally favors short-range antiferromagnetic[41–43] or singlet correlations[36,37]. In such a scenario, the question becomes: how can the coupling of phonons to spins make these phonons chiral (in the presence of a magnetic field)?

## Methods

**Crystal structures.** *La₂CuO₄*: La₂CuO₄ is the parent compound of the most widely studied family of single-layer cuprates, La₂₋ₓSrₓCuO₄. In La₂CuO₄, there is an (apical) oxygen atom above the Cu atom, thereby forming an octahedron of O atoms around Cu (Fig. 1). Upon cooling from high temperature, La₂CuO₄ goes from a tetragonal (I4/mmm) structure to an orthorhombic (Cmca) structure at 530 K (ref. [44]), wherein the octahedra are tilted (the orthorhombic distortion and

associated tilt are not shown in Fig. 1). This means that unless they are deliberately detwinned by application of uniaxial stress, crystals of La₂CuO₄ will be full of orthorhombic structural domains (twins) whose boundaries can in principle scatter phonons. (The samples of La₂CuO₄ studied in refs. [16,21] were twinned.) Below $T_N = 270$ K, the Cu spins order into a collinear antiferromagnetic arrangement, whereby all alternating moments point along the same direction ([110]) within every CuO₂ plane inside a given orthorhombic domain. The tilting of the oxygen octahedra causes a slight canting of the spins out of the CuO₂ plane (by 0.17°) (ref. [45]), thereby producing a Dzyaloshinskii–Moriya (DM) interaction that could, in principle, be a source of chirality.

*Nd₂CuO₄*: Nd₂CuO₄ is the parent compound of the electron-doped family of cuprates Nd₂₋ₓCeₓCuO₄. Unlike La₂CuO₄, it does not undergo any structural transition and remains tetragonal down to $T = 0$. A significant difference from La₂CuO₄ is the absence of apical oxygens, so that Cu atoms in Nd₂CuO₄ are not surrounded by oxygen octahedra (Fig. 1). So, in Nd₂CuO₄, there are no structural domain boundaries and no spin canting.

Magnetically, Nd₂CuO₄ differs from La₂CuO₄ in two ways: there is a large moment on the Nd³⁺ ions and the Cu spins adopt a noncollinear antiferromagnetic order[46]. Below $T_N = 255$ K, the spins of the Cu²⁺ ions order antiferromagnetically along the Cu–O bond ([100]). This breaks the fourfold symmetry within a single CuO₂ plane. However, in the next CuO₂ plane along the c-axis, the same spin configuration is rotated by 90°, thereby restoring the fourfold symmetry of the entire system. This noncollinear magnetic structure therefore preserves the tetragonal symmetry of the crystal.

*Sr₂CuO₂Cl₂*: Sr₂CuO₂Cl₂ has the same crystal structure as tetragonal La₂CuO₄ ($T > 530$ K), with La replaced by Sr and the apical O replaced by Cl (Fig. 1). Unlike La₂CuO₄, it remains tetragonal down to low temperature and its octahedra show no sign of tilting[47–49]. So here, again, there are no structural domain boundaries and no spin canting. Sr₂CuO₂Cl₂ develops collinear antiferromagnetic order below $T_N = 250$ K, with a magnetic structure similar to that of La₂CuO₄ (moments along [110]), except with no spin canting out of the plane[48]. It remains in the same magnetic phase down to at least $T = 10$ K (ref. [48]).

**Samples.** Our single crystal of Nd₂CuO₄ was grown at the University of Science and Technology of China by a standard flux method, annealed in helium for 10 h at 900 °C, and cut in the shape of rectangular platelets with dimensions $0.50 \times 0.69 \times 0.066$ mm³ (length between contacts × width × thickness in the c direction). Contacts were made with silver epoxy, diffused at 500 °C for 1 h. The thermal conductivity $\kappa_{xx}$ of similar samples was studied in detail at low temperature ($T < 20$ K) (refs. [50,51]). Single crystals of Sr₂CuO₂Cl₂ were grown at the University of British Columbia using a flux-growth method. Here, we report data on two samples (labeled A and B), cut in the shape of rectangular platelets with dimensions $0.6 \times 0.11 \times 0.03$ mm³. Contacts were made using silver paint. In all cases, the heat current was made to flow along the a-axis of the tetragonal structure.

**Measurements.** The thermal conductivity $\kappa_{xx}$ was measured applying a heat current $J_x$ within the CuO₂ plane, generating a longitudinal temperature difference $\Delta T_x = T^+ - T^-$. The thermal conductivity along the x-axis is given by $\kappa_{xx} = (J_x/\Delta T_x)(L/wt)$, where $L$ is the distance between $T^+$ and $T^-$, $w$ is the width of the sample, and $t$ its thickness. By applying a magnetic field along the c-axis of the crystal, normal to the CuO₂ planes, a transverse temperature gradient, $\Delta T_y$, was generated (see inset of Supplementary Fig. 1). The thermal Hall conductivity is defined as

$$\kappa_{xy} = -\kappa_{yy}(\Delta T_x/\Delta T_y)(L/w)$$

where $\kappa_{yy}$ is the longitudinal thermal conductivity along the y-axis. Due to the tetragonal structure of our samples, we can take $\kappa_{xx} = \kappa_{yy}$.

The measurements were made with a steady-state method as a function of temperature, using differential type-E thermocouples for $\Delta T_x$ and $\Delta T_y$ (see inset of Supplementary Fig. 1). This method consists in keeping the sample in a fixed magnetic field $H$ and changing its temperature in discrete steps, typically of 2–3 K. At each fixed temperature, the background value of the thermocouple that measures $\Delta T_y$ is recorded before sending heat $J$ to the sample. Once the sample is entirely in equilibrium, we measure $\Delta T_y(H)$. Here, the voltage (heat-off) background in the thermocouple is carefully subtracted from the heat-on signal of the thermocouple to give the correct $\Delta T_y(H)$. Once the entire temperature range is covered, say from 10 to 100 K, the field direction is reversed to $-H$. The same procedure is now applied for these negative values of the field. We then define $\Delta T_y(H) = [\Delta T_y(T,H) - \Delta T_y(T, -H)]/2$, thereby removing any symmetric contamination of the signal coming from the longitudinal thermal gradient and the possible misalignment of transverse contacts.

The heat current along the x-axis is generated by a heater stuck at one end of the sample. The other end is glued to a block that serves as a heat sink (see inset of Supplementary Fig. 1). For the data reported here (and in refs. [16,21]), this block was made of copper. To confirm that the Hall response of copper in a field does not contaminate the Hall response coming from the sample, we performed the same measurement twice, once with the copper block (using metallic contacts made with Ag paint) and then with a block made of the insulator LiF (using insulating contacts made with GE varnish), using the same sample of Nd₂CuO₄ in both cases. The results are shown in Supplementary Fig. 1; we see that the same $\kappa_{xy}$ curve is

obtained with the two setups. We conclude that using copper for the heat sink does not lead to any detectable contamination of the thermal Hall signal.

## Data availability

The data that support the findings of this study are available from the corresponding author upon reasonable request. Source data are provided with this paper.

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

## Acknowledgements
We thank L. Balents, K. Behnia, R.M. Fernandes, B. Flebus, I. Garate, J.H. Han, C. Hess, S.A. Kivelson, P.A. Lee, A.H. MacDonald, J.E. Moore, B.J. Ramshaw, L. Savary, O. Sushkov, A.-M.S. Tremblay, R. Valenti, and C.M. Varma for the fruitful discussions. We thank S. Fortier for his assistance with the experiments. L.T. acknowledges support from the Canadian Institute for Advanced Research (CIFAR) as a CIFAR Fellow and funding from the Natural Sciences and Engineering Research Council of Canada (NSERC, PIN: 123817), the Fonds de recherche du Québec—Nature et Technologies (FRQNT), the Canada Foundation for Innovation (CFI), and a Canada Research Chair. This research was undertaken, thanks in part to funding from the Canada Research Excellence Fund. Part of this work was funded by the Gordon and Betty Moore Foundation's EPiQS Initiative (Grant GBMF5306 to L.T.). This work was also supported by the National Natural Science Foundation of China (11888101 and 11534010).

## Author contributions
M.-E.B., G.G., S.B., A.A., E.L., A.L., and A.G. performed the thermal Hall measurements. M.-E.B., G.G., S.B., A.A., E.L., and A.L. analyzed the results. M.D. performed the X-ray diffraction measurements. C.H.W. and X.H.C. grew the $Nd_2CuO_4$ crystal. R.L., W.N.H., and D.A.B. grew the $Sr_2CuO_2Cl_2$ crystals. M.-E.B., G.G., and L.T. wrote the paper, in consultation with all authors. L.T. supervised the project.

## Competing interests
The authors declare no competing interests.
