## [Peer Review File · Nature Communications]

REVIEWER COMMENTS

Reviewer #1 (Remarks to the Author):

The parent compound of cuprates has been generally believed to be a Mott insulator without low-energy charge excitations. The nature of the Mott state, with possibly nontrivial topological excitations, is a long-standing issue over past three decades. Hence the recent observation of large thermal Hall signals is a surprising breakthrough and has attracted intensive debates concerning its origin. It was first attributed to gapless spinons, but various scenarios such as magnons, majorana fermions and phonons have also been proposed. It is therefore extremely important to clarify these different mechanisms before a decisive conclusion can be drawn.

The current paper is a more elaborated study of the authors' previous discovery and may help to settle above issues. The new results have excluded the contributions of structural orthorhombicity, apical oxygens, the tilting of oxygen octahedra, as well as the canting of spins out of the CuO_2 planes. The authors have suggested possible mechanism based on chiral phonons, which I think is likely. The suppression of κ_{xy}/T^3 and the difference between longitudinal and transverse thermal conductivities seem to not only exclude the possibility of gapless spinons but also suggests a large power-law exponent in the energy dependence of the Berry curvature. The latter is most probably from phonons, the chirality of which is confirmed by the excellent fit to the exponential scaling of the high-temperature data of κ_{xy}/T and could potentially come from the coupling with a nontrivial topological background. The current work clarifies some of the candidate scenarios and points to a future direction for the development of a coupling/decoupling mechanism of phonons. This is an important advance for the understanding of the issue. I therefore recommend its publication in the current form.

Reviewer #2 (Remarks to the Author):

In this manuscript, Boulanger et al. presented a comparative study of thermal Hall conductivity in two cuprate Mott insulators Nd_2CuO_4 and $\text{Sr}_2\text{CuO}_2\text{Cl}_2$, with significantly different crystal structures and magnetic orders and suggested a mechanism underlying the mystery of unexpectedly large thermal Hall conductivity in cuprate Mott insulators: coupling of phonons to the intrinsic excitations of the CuO_2 plane.

Overall, the manuscript was very well-written, and the analysis was compelling to rule out the mechanism of scattering off rare-earth impurities, structural domain walls and the mechanism of spin canting. This is of great importance to the field and suggests that phonon Hall effects in cuprate Mott insulators can be very different from other known cases. The underlying physics might be completely new to the field of condensed matter physics and could potentially be a fundamental breakthrough.

For this reason, I believe the present manuscript should make significant contribution to the studies of thermal Hall effect and will be of great interest to readers of Nature Communications. However, I do have several questions that might need clarifications from the authors, especially on the conclusion that the large thermal Hall conductivity is due to the coupling of phonons to excitations of CuO_2 plane.

1. As the authors mentioned that in LCO, the thermal Hall conductivity κ_{xy} is roughly equal to κ_{zy} , is it possible to speculate what happens here such that the in-plane excitations produce a very small anisotropy? Also, out of curiosity, I am wondering how the κ_{zy} values of the Nd_2CuO_4 and $\text{Sr}_2\text{CuO}_2\text{Cl}_2$ behave in these experiments.

2. I am also curious about whether there is an anisotropy in as one rotates the temperature gradient direction in x-y plane and measures thermal current in z direction. I am inclined to think that if excitations that interact with phonons are from spins in the CuO₂ plane, the orientation and wavevector of the antiferromagnet order parameter in these Mott insulators will enter the game and produce an anisotropic signature.

3. I agree that structural domain walls and rare earth defects do not play a role in the mechanism of thermal Hall according to the experimental results. However, I am a little concerned about the argument from line 280 to 287 that argues that other charged defects such as oxygen vacancies, do not enter the mechanism. These charged defects might be still there, but the effect of them on phonon might change greatly due to the change of electronic states, i.e. the charge might be screened when we approach the metallic phase.

4. A typo on line 281, $p=21$ should be $p=0.21$.

Reviewer #3 (Remarks to the Author):

This work by Boulanger et.al. belongs to a series of experiments on the thermal transport in cuprates. These experiments are probably the most striking discoveries on cuprates in recent years, especially the large thermal transport signals in the undoped Mott insulator phase of the materials. The current work has extended the previous measurements to several new materials that belong the same family, and further consolidated the previous discoveries.

I think this work deserves publication on Nat. Comm. But I am not sure why the authors did not measure κ_{zy} on Nd₂CuO₄ and Sr₂CuO₂Cl₂ like on La₂CuO₄, which would have made the story more complete. In La₂CuO₄ the observation that $\kappa_{zy} \sim \kappa_{xy}$ was one of the main arguments for the "phonon chirality".

But other than this, I would recommend publication of this work.

Detailed response to referee reports

Reviewer #1

Comment:

The parent compound of cuprates has been generally believed to be a Mott insulator without low-energy charge excitations. The nature of the Mott state, with possibly nontrivial topological excitations, is a long-standing issue over past three decades. Hence the recent observation of large thermal Hall signals is a surprising breakthrough and has attracted intensive debates concerning its origin. It was first attributed to gapless spinons, but various scenarios such as magnons, majorana fermions and phonons have also been proposed. It is therefore extremely important to clarify these different mechanisms before a decisive conclusion can be drawn.

The current paper is a more elaborated study of the authors' previous discovery and may help to settle above issues. The new results have excluded the contributions of structural orthorhombicity, apical oxygens, the tilting of oxygen octahedra, as well as the canting of spins out of the CuO_2 planes. The authors have suggested possible mechanism based on chiral phonons, which I think is likely. The suppression of κ_{xy}/T^3 and the difference between longitudinal and transverse thermal conductivities seem to not only exclude the possibility of gapless spinons but also suggests a large power-law exponent in the energy dependence of the Berry curvature. The latter is most probably from phonons, the chirality of which is confirmed by the excellent fit to the exponential scaling of the high-temperature data of κ_{xy}/T and could potentially come from the coupling with a nontrivial topological background. The current work clarifies some of the candidate scenarios and points to a future direction for the development of a coupling/decoupling mechanism of phonons. This is an important advance for the understanding of the issue. I therefore recommend its publication in the current form.

Our response:

We thank the referee for his/her appreciation of our work and for recommending publication.

Reviewer #2

Comment:

In this manuscript, Boulanger et al. presented a comparative study of thermal Hall conductivity in two cuprate Mott insulators Nd_2CuO_4 and $\text{Sr}_2\text{CuO}_2\text{Cl}_2$, with significantly different crystal structures and magnetic orders and suggested a mechanism underlying the mystery of unexpectedly large thermal Hall conductivity in cuprate Mott insulators: coupling of phonons to the intrinsic excitations of the CuO_2 plane.

Overall, the manuscript was very well written, and the analysis was compelling to rule out the mechanism of scattering off rare-earth impurities, structural domain walls and the mechanism of spin canting. This is of great importance to the field and suggests that phonon Hall effects in cuprate Mott insulators can be very different from other known cases. The underlying physics might be completely new to the field of condensed matter physics and could potentially be a fundamental breakthrough.

For this reason, I believe the present manuscript should make significant contribution to the studies of thermal Hall effect and will be of great interest to readers of Nature Communications.

Our response:

We thank the referee for his/her appreciation of our work and for recommending publication.

Comments :

However, I do have several questions that might need clarifications from the authors, especially on the conclusion that the large thermal Hall conductivity is due to the coupling of phonons to excitations of CuO₂ plane.

1. As the authors mentioned that in LCO, the thermal Hall conductivity κ_{xy} is roughly equal to κ_{zy} , is it possible to **speculate** what happens here such that the in-plane excitations produce a very small anisotropy? Also, **out of curiosity**, I am wondering how the κ_{zy} values of the Nd₂CuO₄ and Sr₂CuO₂Cl₂ behave in these experiments.

Our response:

We have not measured κ_{zy} in Nd₂CuO₄ or Sr₂CuO₂Cl₂. This is something we plan to do, but it requires samples that are thick in the c axis direction, which we do not currently have. This is clearly an important future project.

As for the small anisotropy between κ_{zy} and κ_{xy} in La₂CuO₄, reported in our preceding paper (Nature Physics 2020), it is indeed intriguing. We can only mention that phonons scattered by in-plane electrons at non-zero doping also yield a very small anisotropy between κ_{zz} and (the phonon part) of κ_{xx} . This is shown in Figure A below for Nd-LSCO at $p = 0.21$ and $p = 0.24$.

These data show that although electronic excitations are strongly 2D, confined to the CuO₂ planes, their coupling to phonons does not cause a pronounced *a-c* anisotropy of the phonon conductivity.

[Redacted]

Figure A | Longitudinal thermal conductivity of Nd-LSCO at two dopings: $p = 0.21$ (left) and $p = 0.24$ (right). The c axis conductivity $\kappa_{zz} = \kappa_c$ is plotted in blue, as κ_{zz}/T vs T . It is purely phononic. We see that κ_{zz} at $T \sim 10$ K increases by a factor ~ 3 upon entering the pseudogap phase. This is because the charge carrier density drops and hence the electron scattering of phonons weakens. In red, we plot the in-plane conductivity of phonons, estimated as the difference between the measured κ_{xx}/T and the electronic contribution estimated from the Wiedemann-Franz law, $L_0\sigma_{xx}$. We see from the data at $p = 0.24$ (right), that there is only a small anisotropy between in-plane and out-of-plane phonon transport (ratio ~ 1.6 at 10 K), even though at $T \sim 10$ K the dominant scattering mechanism of phonons is electron scattering and these are confined to the CuO₂ planes.

Comment:

2. I am also curious about whether there is an anisotropy in as one rotates the temperature gradient direction in x-y plane and measures thermal current in z direction. I am inclined to think that if excitations that interact with phonons are from spins in the CuO_2 plane, the orientation and wavevector of the antiferromagnet order parameter in these Mott insulators will enter the game and produce an anisotropic signature.

Our response:

Again, we have not measured κ_{zy} (or κ_{zx}) in Nd_2CuO_4 or $\text{Sr}_2\text{CuO}_2\text{Cl}_2$. Once we have acquired the necessary samples (thick along the c-axis direction), we will certainly consider the referee's interesting proposal. In particular in samples for which a single AF domain has been created, using the field-cooling method described in our manuscript.

Comment:

3. I agree that structural domain walls and rare earth defects do not play a role in the mechanism of thermal Hall according to the experimental results. However, I am a little concerned about the argument from line 280 to 287 that argues that other charged defects such as oxygen vacancies, do not enter the mechanism. These charged defects might be still there, but the effect of them on phonon might change greatly due to the change of electronic states, i.e. the charge might be screened when we approach the metallic phase.

Our response:

The possibility that phonons acquire chirality because they are scattered by charged impurities such as oxygen vacancies is an interesting one. We argue that this mechanism is ruled out by the fact that κ_{zy} is zero in Nd-LSCO at $p = 0.24$ (and also in Eu-LSCO at $p = 0.24$).

We agree with the referee that the charged impurities would be screened by mobile electrons more effectively at $p = 0.24$ (high carrier density) than at $p = 0.21$ (low carrier density). But this should be a quantitative difference, related to the resistivity of each sample, for example. So we would expect to see some non-negligible κ_{zy} signal at $p = 0.24$. It would be smaller than at $p = 0.21$, but it would not be zero.

In this scenario of screened oxygen vacancies, we would also expect zero screening at $p = 0$, in the insulator, so we should see a much larger κ_{zy} signal in LCO than in Nd-LSCO at $p = 0.21$, for example. This is not the case, on the contrary. As seen from Table 1, the κ_{zy} signal is *larger* at $p = 0.21$, relative to κ_{zz} : $|\kappa_{zy} / \kappa_{zz}| = 0.48\%$ in Nd-LSCO $p = 0.21$ vs $|\kappa_{zy} / \kappa_{zz}| = 0.2\%$ in LCO.

***** In the revised manuscript, we have added a sentence emphasizing these points.**

Comment:

4. A typo on line 281, $p = 21$ should be $p = 0.21$.

Our response:

This has now been corrected.

Reviewer #3

Comment:

This work by Boulanger *et.al.* belongs to a series of experiments on the thermal transport in cuprates. These experiments are probably the most striking discoveries on cuprates in recent years, especially the large thermal transport signals in the undoped Mott insulator phase of the materials. The current work has extended the previous measurements to several new materials that belong to the same family, and further consolidated the previous discoveries.

Our response:

We thank the referee for his/her appreciation of our work.

Comments:

I think this work deserves publication on Nat. Comm. But I am not sure why the authors did not measure κ_{zy} on Nd_2CuO_4 and $\text{Sr}_2\text{CuO}_2\text{Cl}_2$ like on La_2CuO_4 , which would have made the story more complete. In La_2CuO_4 the observation that $\kappa_{zy} \sim \kappa_{xy}$ was one of the main arguments for the "phonon chirality".

But other than this, I would recommend publication of this work.

Our response:

We thank the referee for recommending publication.

As mentioned above, we are planning to measure κ_{zy} in future experiments. This will require samples that are thick in the c-axis direction, which we do not currently have.

As emphasized in our manuscript, we are providing a different, and also strong, argument in favor of phonon chirality: the fact that the magnitude of κ_{xy} in the three different cuprate Mott insulators tracks the magnitude of the phonon-dominated κ_{xx} .

REVIEWERS' COMMENTS

Reviewer #1 (Remarks to the Author):

I read the response and think the authors have addressed all the issues in the referees' reports. I have no further comment.

Reviewer #2 (Remarks to the Author):

I appreciate the response and the corresponding revision of the manuscript. The authors have appropriately revised the manuscript and satisfactorily addressed my concerns in the first report. I believe this work should make a significant contribution in the study of thermal Hall transport and therefore recommend publication of the present manuscript.

Reviewer #3 (Remarks to the Author):

As far as I see the authors have addressed my comments properly. I recommend the manuscript for publication on Nat. Comm.

Detailed response to referee reports

Reviewer #1

Comment:

I read the response and think the authors have addressed all the issues in the referees' reports. I have no further comment.

Our response:

We thank the referee for his / her appreciation of our work and for recommending publication.

Reviewer #2

Comment:

I appreciate the response and the corresponding revision of the manuscript. The authors have appropriately revised the manuscript and satisfactorily addressed my concerns in the first report. I believe this work should make a significant contribution in the study of thermal Hall transport and therefore recommend publication of the present manuscript.

Our response:

We thank the referee for his / her appreciation of our work and for recommending publication.

Reviewer #3

Comment:

As far as I see the authors have addressed my comments properly. I recommend the manuscript for publication on Nat. Comm.

Our response:

We thank the referee for his/her appreciation of our work for recommending publication.